# Healthcare utilization, mortality, and cardiovascular events following GLP1-RA initiation in chronic kidney disease

Shuyao Zhang[1], Fnu Sidra[1,2], Carlos A. Alvarez[3], Mustafa Kinaan[4,5], Ildiko Lingvay [1,6,8] & Ishak A. Mansi [5,7,8] ✉

Treatment with glucagon-like peptide-1 receptor agonists (GLP1-RA) in patients with type 2 diabetes (T2D) and chronic kidney disease (CKD) may attenuate kidney disease progression and cardiovascular events but their real-world impact on healthcare utilization and mortality in this population are not well-defined. Here, we emulate a clinical trial that compares outcomes following initiation of GLP1-RA vs Dipeptidyl peptidase-4 inhibitors (DPP4i), as active comparators, in U.S. veterans aged 35 years of older with moderate to advanced CKD during fiscal years 2006 to 2021. Primary outcome was rate of acute healthcare utilization. Secondary outcomes were all-cause mortality and a composite of acute cardiovascular events. After propensity score matching (16,076 pairs) and 2.2 years mean follow-up duration, use of GLP1-RA in patients with moderate to advanced CKD was associated with lower annual rate of acute healthcare utilization and all-cause mortality. There was no significant difference in acute cardiovascular events.

Diabetic kidney disease is the leading cause of end stage renal disease (ESRD) and results in considerable economic burden[1–3]. Patients with chronic kidney disease (CKD) and ESRD due to type 2 diabetes (T2D) are prone to complications such as frequent hypoglycemia, infections, or cardiovascular events, which lead to increased healthcare utilization[4–7]. Adjusted rates of hospitalization for patients with CKD are consistently higher than similar patients without CKD, and worse CKD stage are associated with increased rate of hospitalization[8].

Historically, glucose lowering agents approved for management of T2D in patients with advanced CKD have been limited to insulin and sulfonylureas despite their associated risk of hypoglycemia and neutral effect on cardiovascular outcomes or progression of kidney disease[9]. Dipeptidyl peptidase-4 inhibitors (DPP4i) and Glucagon Like Peptide-1 Receptor Agonists (GLP1-RA) were introduced around 2006 and are now increasingly prescribed due to their demonstrated safety

in advanced kidney disease[10–12]. Both the Kidney Disease Improving Global Outcomes guidelines and American Diabetes Association Standards of Care recommend GLP1-RA as the preferred glucose lowering class for people with T2D and moderate to advanced CKD or ESRD because of their demonstrated cardiovascular benefits in people with T2D at high risk of cardiovascular disease, which includes CKD[13,14].

Despite these recommendations, there is limited real-world data evaluating outcomes of GLP1-RAs specifically in patients with moderate to advanced CKD or ESRD. Small-scale studies in patients on hemodialysis show promising results with decreased risk of hypoglycemia, reduction in insulin dose requirement, and improved glycemic control with use of GLP1-RA[15,16]. However, data to the contrary also exist and some small studies raised concerns that GLP1-RA use in CKD may be associated with more gastrointestinal side effects, hypoglycemic events, and loss of muscle mass[17,18]. There is little published data

[1]Department of Internal Medicine, Division of Endocrinology, University of Texas Southwestern Medical Center, Dallas, TX, USA. [2]The Jones Center for Diabetes & Endocrine Wellness, Macon, GA, USA. [3]Department of Pharmacy Practice and Center for Excellence in Real World Evidence, Texas Tech University Health Science Center, Dallas, TX, USA. [4]Endocrinology, Diabetes, and Metabolism Fellowship, UCF HCA Healthcare GME, Orlando, FL, USA. [5]Department of Internal Medicine, University of Central Florida, College of Medicine, Orlando, FL, USA. [6]Peter O'Donnell Jr. School of Public Health, University of Texas Southwestern Medical Center, Dallas, TX, USA. [7]Education Services, Orlando VA Healthcare System, Orlando, FL, USA. [8]These authors contributed equally: Ildiko Lingvay, Ishak A. Mansi. ✉e-mail: Ishak.mansi@va.gov

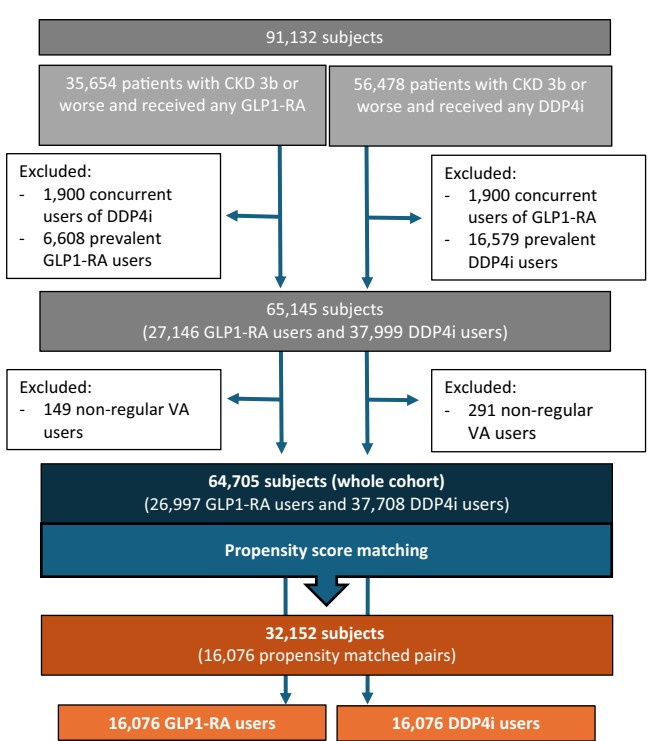

**Fig. 1 | Flow chart of patient selection and final study sample.** CKD chronic kidney disease, GLP1-RA glucagon like peptide 1 receptor agonists, DPP4i dipeptidyl peptidase 4 inhibitors, VA veteran affairs.

on the effect of GLP1-RA therapy on American healthcare resource utilization in patients with CKD. Furthermore, GLP1-RA is known to be associated with decreased all-cause mortality and cardiovascular mortality in the general diabetes population[19–21], but real-world data in patients with moderate to advanced kidney disease is limited.

In this work, we emulated a target clinical trial comparing rate of acute healthcare utilization, all-cause mortality, composite cardiovascular events, and kidney disease progression in patients with moderate to advanced CKD initiating GLP1-RAs versus DPP4is in real world practice within the Veterans Healthcare Administration (VHA).

## Results
### Baseline patient characteristics
In total, 92,132 patients with CKD stage 3b or worse who filled prescriptions for GLP1-RA or DPP4i were identified. After applying inclusion and exclusion criteria, 64,705 patients (26,997 GLP1-RA users and 37,708 DPP4i users) were included in the analysis (Fig. 1). GLP1-RA users used GLP1-RA medications for a mean (SD) of 568 (532) days. DPP4i users used DPP4i medications for 772 (722) days. Among GLP1-RA users, 8315 (30.8%) patients discontinued GLP1-RA after a mean (SD) period of 434 (490) days. Among DPP4i users, 17,658 (46.8%) discontinued DPP4i after a mean (SD) period of 603 (634) days. Among GLP1-RA users, 6727 (24.9%) initially used DPP4i before switching to GLP1-RA. Among DPP4i users, 819 (2.2%) initially used GLP1-RA before switching to DPP4i. These data are shown in Supplementary Fig. S1 and the full list of baseline characteristics of the unmatched cohort is outlined in Table 1.

After propensity score matching, 16,076 matched pairs of GLP1-RA and DPP4i users were identified and included in the primary analysis. The propensity score distribution between the groups was balanced (Supplemental Fig. S2). After propensity matching, there was no significant difference in baseline characteristics between the two groups for all variables included. The full list of baseline characteristics by propensity matched group is outlined in Table 1. In the propensity

score-matched cohort, the mean age of the matched patients was 72 years of age, 95% were male with mean BMI of 33.5 (SD 5.9) kg per m². New prescription initiation increased each interval year, with the highest number of new prescriptions prescribed between FY 2019–2021 (66.3% of GLP1-RA and 67% of DPP4i initiations). At baseline, 98.7% of patients in both groups had HbA1c > 6.5%, with mean HbA1c of 8.0% (SD 1.1 in GLP1-RA group and 1.2 DPP4i, respectively) and mean glucose of 172 (SD 35 and 37) mg per dL. On average both groups used 2.6 (SD 1.2 and 1.1) anti-diabetes medication classes and 73.6% and 74.9% of patients used insulin during the baseline period in the GLP1-RA and DPP4i groups, respectively. Most patients had diabetes with complications: 66.5% and 66.9% of patients had diabetes ketoacidosis or uncontrolled diabetes, 15.1% and 15% had documented hypoglycemia within the GLP1-RA and DPP4i groups, respectively. Mean baseline eGFR was 48.5 (SD 11.7 and 11.8) mL per min per 1.73 m². Cardiac comorbidities at baseline were highly prevalent: 98.6% and 98.5% of patients had hypertension, 55.2% and 55.1% of patients had coronary artery disease, and 33.9% and 34% of patients had congestive heart failure, within the GLP1-RA and DPP4i groups, respectively. Both GLP1-RA and DPP4i groups had an average of 2.3 (SD 4.4 and 4.1) inpatient admissions and 7.8 (14.1 and 12.9) ED visits during the baseline period which had an average duration of 132 months (-11 years) in both groups. The annual rate of acute healthcare visits was 0.93 and 0.94 events/year, and use of durable medical equipment was 40.2% and 40.9% for the GLP1-RA and DPP4i groups, respectively.

### Primary and secondary outcomes in propensity-matched cohort
The rates of acute healthcare utilization, all-cause mortality, combined cardiovascular composite event outcome, and CKD progression of the two study groups are shown in Table 2A. After propensity score matching, the annual rate of acute healthcare utilization was lower within the GLP1-RA group compared to the propensity matched DPP4i group (Table 2). During an average follow-up of 26.3 months (2.2 years, SD 1.9 years), the annual rate of acute healthcare utilization was 1.52 visits/year in the GLP1-RA group and 1.67 visits/year within the DPP4i group (coefficient of regression $\beta = -0.15$, 95% CI −0.25 to −0.05, $p = 0.004$).

All-cause death during the follow-up period was significantly lower within the GLP1-RA group compared to the DPP4i group, with 2847 (17.7%) deaths within GLP1-RA group compared to 3287 (20.5%) in DPP4 group (OR 0.84, 95% CI 0.79 to 0.89, $p < 0.01$). The incidence of the cardiovascular composite event outcome was not significantly different between groups. In the GLP1-RA group 1757 (10.9%) people experienced an event, compared to 1782 (11.1%) within the DPP4i group (OR 0.98, 95% CI 0.92 to 1.06, $p = 0.66$). Time to all-cause death analysis showed a HR of 0.86 (95%CI: 0.82–0.90, $p < 0.001$), and the time to first event in the cardiovascular composite event outcome showed a HR of 0.99 (95%CI: 0.93–1.06, $p = 0.82$) (Fig. 2).

### Post-hoc analyses
CKD progression composite outcome was decreased within the GLP1-RA group as compared to DPP4i group, with 2.23% of patients within GLP1-RA group having significant progression of CKD as compared to 3.46% within the DPP4i group (OR 0.64., 95% CI 0.56–0.73, $p < 0.001$). In a post-hoc analysis of microalbuminuria during follow-up, GLP1-RA users has significantly lower levels compared with DPP4i users; however, this analysis is limited by a large proportion of data missingness (Supplementary Table S1).The pattern of association of GLP1-RA with decreased acute healthcare utilization, CKD progression and all-cause death despite no difference in CV outcomes persisted in additional analyses excluding patients with ESRD (Supplementary Table S2) and after adjusting for A1c at follow-up and use of other anti-hyperglycemic medications (Supplementary Table S3).

A post-hoc analysis of selected safety outcomes showed increased odds of hypoglycemic events in propensity score-matched GLP1-RA users compared to DPP4i users (10.3% vs 9.3%, respectively; OR 1.13,

**Table 1 | Baseline characteristics of GLP1-RA and DPP4i groups before and after propensity score-matching**

| | PS-matched cohort | | | Cohort before matching | | |
|---|---|---|---|---|---|---|
| | GLP1-RA (n = 16,076) | DPP4i (n = 16,076) | St Diff | GLP1-RA (n = 26,997) | DPP4i (n = 37,708) | St Diff |
| Baseline demographics and characteristics | | | | | | |
| Age, years | 71.9 (8.1) | 71.8 (8.7) | 0.01 | 71.1 (8.0) | 73.9 (9.4) | 0.330 |
| Male Gender | 15,344 (95) | 15,344 (95) | <0.001 | 25,697 (95) | 36,243 (96) | 0.046 |
| Race | | | 0.013 | | | 0.045 |
| Black | 3277 (20.4) | 3700 (23) | | 5349 (19.8) | 8360 (22.2) | |
| White | 11,511 (71.6) | 10,861 (67.6) | | 19,502 (72.2) | 25,582 (67.8) | |
| Other races | 472 (2.9) | 447 (2.8) | | 698 (2.6) | 1268 (3.4) | |
| Unknown/missing | 816 (5.1) | 1068 (6.6) | | 1448 (5.4) | 2498 (6.6) | |
| Study intervals | | | | | | |
| Index date period | | | | | | |
| Years 2006–2009 | 124 (0.8) | 112 (0.7) | 0.009 | 161 (0.6) | 687 (1.8) | 0.112 |
| Years 2010–2012 | 239 (1.5) | 243 (1.5) | 0.002 | 259 (1.0) | 2072 (5.5) | 0.027 |
| Years 2013–2015 | 693 (4.3) | 605 (3.8) | 0.028 | 717 (2.7) | 5194 (13.8) | 0.413 |
| Years 2016–2018 | 4355 (27) | 4339 (27) | 0.002 | 5790 (21.5) | 12,287 (32.6) | 0.253 |
| Years 2019-2021 | 10,665 (66.3) | 10,777 (67) | 0.015 | 20,070 (74.3) | 17,468 (46.3) | 0.598 |
| Duration of baseline period, days | 3950 (1505) | 3953 (1449) | 0.002 | 4051 (1471) | 3597 (1497) | 0.307 |
| Duration of follow up period, days | 795 (701) | 789 (691) | 0.008 | 727 (637) | 1031 (860) | 0.402 |
| Social and family history during baseline period | | | | | | |
| Family history of cardiovascular diseases | 733 (4.6) | 720 (4.5) | 0.004 | 1401 (5.2) | 1374 (3.6) | 0.075 |
| Smoking at any time | 6419 (39.9) | 6459 (40.2) | 0.005 | 11,219 (41.6) | 13,697 (36.3) | 0.107 |
| Alcohol-related disorders | 1615 (10.1) | 1615 (10.1) | 0 | 2657 (9.8) | 3459 (9.2) | 0.023 |
| Substance-related disorders | 2090 (13) | 2105 (13.1) | 0.003 | 3701 (13.7) | 2990 (10.6) | 0.096 |
| Vital signs during baseline period | | | | | | |
| Systolic blood pressure, mmHg | 137 (11) | 137 (11) | 0.006 | 136 (11) | 137 (11) | 0.033 |
| Diastolic blood pressure, mmHg | 75 (7) | 75 (7) | 0.005 | 75 (7) | 75 (7.6) | 0.073 |
| Body mass index, kg per m | 33.5 (5.9) | 33.5 (5.9) | 0.004 | 34.6 (6.1) | 31.8 (5.7) | 0.478 |
| <25 kg per m$^2$ | 677 (4.2) | 652 (4.1) | 0.008 | 776 (2.9) | 3339 (8.9) | 0.256 |
| 25 to <30 kg per m$^2$ | 4171 (26.0) | 4114 (25.6) | 0.008 | 5560 (20.6) | 12,640 (33.5) | 0.294 |
| 30 to <35 kg per m$^2$ | 5691 (35.5) | 5788 (36.0) | 0.013 | 9291 (34.4) | 12,472 (33.1) | 0.028 |
| 35 to <40 kg per m$^2$ | 3413 (21.2) | 3406 (21.2) | 0.001 | 6706 (24.8) | 6066 (16.1) | 0.218 |
| 40 to <45 kg per m$^2$ | 1439 (9.0) | 1427 (8.9) | 0.003 | 3063 (11.4) | 2198 (5.8) | 0.198 |
| ≥45 kg per m$^2$ | 685 (4.3) | 689 (4.3) | 0.001 | 1601 (5.9) | 993 (2.6) | 0.163 |
| Healthcare utilization during baseline period | | | | | | |
| Number of outpatient encounters | 314 (301) | 316 (318) | 0.005 | 348 (318) | 251 (279) | 0.327 |
| Number of inpatient admissions | 2.3 (4.4) | 2.3 (4.1) | <0.001 | 2.5 (4.4) | 1.9 (3.8) | 0.145 |
| Number of ED visits | 7.8 (14.1) | 7.8 (12.9) | 0.002 | 8.3 (13.8) | 6.3 (12.3) | 0.153 |
| Number of combined ED and hospitalization | 10.1 (17.5) | 10.1 (16.0) | 0.006 | 10.8 (17.1) | 8.2 (15.1) | 0.012 |
| Annual rate of acute healthcare utilization, EPY | 0.93 (1.9) | 0.94 (1.9) | 0.006 | 0.97 (1.7) | 0.92 (5.9) | 0.012 |
| Use of durable medical equipment | 6468 (40.2) | 6577 (40.9) | 0.014 | 11,554 (42.8) | 12,854 (34.1) | 0.18 |
| Diabetes and its complications during baseline period | | | | | | |
| Diabetes with ketoacidosis or uncontrolled diabetes | 10,694 (66.5) | 10,749 (66.9) | <0.001 | 19,524 (72.3) | 19,413 (51.5) | 0.439 |
| Diabetes with ophthalmic manifestations | 7676 (47.8) | 7679 (47.8) | <0.001 | 14,171 (52.5) | 14,143 (37.5) | 0.394 |
| Diabetes with neurological manifestations | 9621 (59.9) | 9662 (60.1) | 0.005 | 17,806 (66.0) | 17,618 (46.7) | 0.395 |
| Diabetes with circulatory manifestations | 2321 (14.4) | 2345 (14.6) | 0.004 | 4474 (16.6) | 4020 (10.7) | 0.173 |
| Diabetes with unspecified manifestations | 9190 (57.2) | 9367 (58.3) | 0.022 | 17,532 (64.9) | 14,980 (39.7) | 0.522 |
| Diabetes with hypoglycemia | 2419 (15.1) | 2418 (15.0) | <0.001 | 4600 (17.0) | 4145 (11.0) | 0.175 |
| Any hypoglycemic or hyperglycemic events | 11,063 (68.8) | 11,107 (69.1) | 0.006 | 20,095 (74.4) | 20,417 (54.2) | 0.09 |
| Plasma glucose, mg per dL | 172 (35) | 172 (37) | 0.001 | 177 (356) | 162 (36) | 0.41 |
| Mean HbA1c during baseline period, % | 8.0 (1.1) | 8.0 (1.2) | 0.006 | 8.2 (1.1) | 7.6 (1.1) | 0.52 |
| >6.5% | 15,866 (98.7) | 15,871 (98.7) | 0.003 | 26,763 (99.1) | 36,752 (97.5) | 0.13 |
| >9% | 12,270 (76.3) | 12,419 (77.3) | 0.022 | 22,393 (82.6) | 22,068 (58.5) | 0.55 |
| Glucose lowering medication classes utilized during baseline period | | | | | | |
| Metformin | 11,922 (74.2) | 11,917 (74.1) | 0.001 | 20,817 (77.11) | 25,292 (67.1) | 0.225 |
| Sulphonylurea | 11,895 (74.0) | 11,869 (73.8) | 0.004 | 19710 (73) | 27,943 (74.1) | 0.025 |
| Thiazolidinediones | 4196 (26.1) | 4229 (26.3) | 0.005 | 7407 (27.4) | 9121 (24.2) | 0.074 |

**Table 1 (continued) | Baseline characteristics of GLP1-RA and DPP4i groups before and after propensity score-matching**

| | PS-matched cohort | | | Cohort before matching | | |
|---|---|---|---|---|---|---|
| | GLP1-RA (n = 16,076) | DPP4i (n = 16,076) | St Diff | GLP1-RA (n = 26,997) | DPP4i (n = 37,708) | St Diff |
| α-glucosidase inhibitors | 1097 (6.8) | 1077 (6.7) | 0.005 | 1821 (6.8) | 2616 (6.9) | 0.008 |
| Amylin analog | 36 (0.2) | 31 (0.2) | 0.007 | 101 (0.4) | 36 (0.1) | 0.058 |
| SGLT2i | 1341 (8.3) | 1212 (7.5) | 0.03 | 4199 (15.6) | 1319 (3.5) | 0.421 |
| Insulins | 11,839 (73.6) | 12,036 (74.9) | 0.03 | 22,426 (83.1) | 17,278 (45.8) | 0.845 |
| Number of anti-diabetes medication classes | 2.6 (1.2) | 2.6 (1.1) | 0.001 | 2.8 (1.2) | 2.2 (1.2) | 0.525 |
| Other comorbidities during baseline period | | | | | | |
| Cardiac arrest and ventricular fibrillation | 106 (0.7) | 112 (0.7) | 0.005 | 269 (1) | 184 (0.5) | 0.059 |
| Acute myocardial infarction | 2066 (12.9) | 2019 (12.6) | 0.009 | 4044 (15.0) | 3811 (10.1) | 0.148 |
| Coronary artery disease | 8880 (55.2) | 8854 (55.1) | 0.003 | 16,188 (60.0) | 19,044 (50.5) | 0.191 |
| Acute cerebrovascular disease | 1559 (9.7) | 1543 (9.6) | 0.003 | 2684 (9.9) | 3428 (9.1) | 0.029 |
| Cerebrovascular disease | 4628 (28.8) | 4539 (28.2) | 0.01 | 8063 (29.9) | 10,025 (26.6) | 0.073 |
| Hemiplegia/quadriplegia | 297 (1.9) | 302 (1.9) | 0.002 | 490 (1.8) | 649 (1.7) | 0.007 |
| Dementia | 877 (5.5) | 898 (5.6) | 0.006 | 1266 (4.7) | 2272 (6.0) | 0.059 |
| Congestive heart failure | 5452 (33.9) | 5468 (34.0) | 0.002 | 10,651 (39.5) | 10,510 (27.9) | 0.072 |
| Atrial fibrillation | 6897 (42.9) | 6943 (43.2) | 0.006 | 12305 (45.6) | 14107 (37.4) | 0.166 |
| Hypertension | 15,855 (98.6) | 15,841 (98.5) | 0.007 | 26,680 (98.8) | 36,871 (97.8) | 0.081 |
| PCI procedure | 956 (6.0) | 946 (5.9) | 0.003 | 1846 (6.8) | 1587 (4.2) | 0.115 |
| CABG | 486 (3.0) | 471 (2.9) | 0.005 | 971 (3.6) | 800 (2.1) | 0.089 |
| Peripheral vascular disease | 4958 (30.8) | 4933 (30.7) | 0.005 | 8850 (32.8) | 10389 (27.6) | 0.114 |
| Chronic obstructive pulmonary disease and bronchiectasis | 6485 (40.3) | 6478 (40.3) | 0.001 | 11,671 (43.2) | 13,559 (36.0) | 0.149 |
| Rheumatoid arthritis; Systemic lupus erythematosus and connective tissue disorders | 560 (3.5) | 552 (3.4) | 0.003 | 960 (3.6) | 1172 (3.1) | 0.025 |
| Liver disease-mild | 1356 (8.4) | 1406 (8.8) | 0.011 | 2621 (9.7) | 2461 (6.5) | 0.117 |
| Liver disease-severe | 220 (1.4) | 248 (1.5) | 0.014 | 414 (1.5) | 464 (1.2) | 0.026 |
| Malignancy other than skin cancer | 3781 (23.5) | 3682 (22.9) | 0.01 | 5995 (22.2) | 9263 (24.6) | 0.056 |
| Metastatic neoplasm | 294 (1.8) | 304 (1.9) | 0.005 | 439 (1.6) | 839 (2.2) | 0.044 |
| Acquired Immunodeficiency Syndrome[6] | 115 (0.7) | 107 (0.7) | 0.006 | 179 (0.7) | 253 (0.7) | 0.001 |
| Anemia | 7267 (45.2) | 7271 (45.2) | 0.001 | 12,396 (45.9) | 16,842 (44.7) | 0.025 |
| Thyroid disease | 3149 (19.6) | 3133 (19.5) | 0.003 | 5495 (20.4) | 6881 (18.3) | 0.052 |
| Gait abnormality | 5230 (32.5) | 5233 (32.6) | <0.001 | 9395 (34.8) | 10444 (27.7) | 0.154 |
| Arthritis | 10,250 (63.8) | 10,207 (63.5) | 0.006 | 17,694 (65.5) | 22,606 (60.0) | 0.116 |
| Falls | 2407 (15) | 2426 (15.1) | 0.003 | 4341 (16.1) | 4914 (13) | 0.087 |
| Incontinence | 1851 (11.5) | 1862 (11.6) | 0.002 | 3220 (11.9) | 3881 (10.3) | 0.052 |
| Muscle wasting | 3493 (21.7) | 3505 (21.8) | 0.002 | 6146 (22.8) | 7244 (19.2) | 0.087 |
| Osteoporosis | 611 (3.8) | 605 (3.8) | 0.002 | 963 (3.6) | 1550 (4.1) | 0.028 |
| Parkinsonism | 853 (5.3) | 862 (5.4) | 0.002 | 1481 (5.5) | 1760 (4.7) | 0.037 |
| Peripheral neuropathy | 9523 (59.2) | 9549 (59.4) | 0.003 | 17,705 (65.6) | 17,012 (45.1) | 0.421 |
| Impaired vision | 6747 (42) | 6748 (42) | <0.001 | 11,432 (42.4) | 15,044 (39.9) | 0.05 |
| Weight loss | 1094 (6.8) | 1078 (6.7) | 0.004 | 1606 (6.0) | 2857 (7.6) | 0.065 |
| Anxiety | 4343 (27) | 4340 (27) | <0.001 | 7907 (29.3) | 8544 (22.7) | 0.152 |
| Depression | 7100 (44.2) | 7170 (44.6) | 0.009 | 12,742 (47.2) | 14,049 (37.3) | 0.202 |
| Chronic pain | 5092 (31.7) | 5078 (31.6) | 0.002 | 9321 (34.5) | 9948 (26.4) | 0.178 |
| Failure to thrive | 157 (1) | 164 (1) | 0.004 | 209 (0.8) | 418 (1.1) | 0.035 |
| Fatigue | 4289 (26.7) | 4248 (26.4) | 0.006 | 7841 (29.0) | 8221 (21.8) | 0.167 |
| Hearing loss | 8281 (51.5) | 8265 (51.4) | 0.002 | 14,206 (52.6) | 18,420 (48.9) | 0.075 |
| Obesity | 11,133 (69.3) | 11,170 (69.5) | 0.005 | 20,367 (75.4) | 21,275 (56.4) | 0.41 |
| Comorbidity scores during baseline period | | | | | | |
| Weighted Charlson Comorbidity Total Score[6] | 7.3 (3.3) | 7.3 (3.3) | 0.0005 | 7.6 (3.3) | 6.4 (3.4) | 0.367 |
| Frailty Index | 0.37 (0.15) | 0.37 (0.15) | <0.001 | 0.38 (0.15) | 0.33 (0.15) | 0.325 |
| Non-frail | 273 (1.7) | 285 (1.8) | 0.006 | 348 (1.3) | 1105 (2.9) | 0.114 |
| Pre-frail | 2151 (13.4) | 2137 (13.3) | 0.003 | 2948 (10.9) | 6851 (18.2) | 0.207 |
| Frail-mild | 3775 (23.5) | 3792 (23.6) | 0.002 | 5826 (21.6) | 10,166 (27.0) | 0.126 |
| Frail-moderate | 3865 (24) | 3823 (23.8) | 0.006 | 6512 (24.1) | 8482 (22.5) | 0.039 |
| Frail-severe | 6012 (37.4) | 6039 (37.6) | 0.003 | 11363 (42.1) | 11104 (29.5) | 0.267 |

**Table 1 (continued) | Baseline characteristics of GLP1-RA and DPP4i groups before and after propensity score-matching**

| | PS-matched cohort | | | Cohort before matching | | |
|---|---|---|---|---|---|---|
| | GLP1-RA (n = 16,076) | DPP4i (n = 16,076) | St Diff | GLP1-RA (n = 26,997) | DPP4i (n = 37,708) | St Diff |
| Cardiovascular risk | 20.7 (5.3) | 20.1 (5.3) | 0.004 | 20.6 (5.3) | 21.0 (5.3) | 0.09 |
| <5% | 70 (0.4) | 71 (0.4) | 0.001 | 109 (0.4) | 160 (0.4) | 0.003 |
| 5 to <10% | 660 (4.1) | 686 (4.3) | 0.008 | 1201 (4.5) | 1288 (3.4) | 0.053 |
| 10 to <15% | 1462 (9.1) | 1447 (9) | 0.003 | 2445 (9.1) | 3194 (8.5) | 0.021 |
| 15 to <20% | 4399 (27.4) | 4425 (27.5) | 0.004 | 7547 (28.0) | 9744 (25.8) | 0.048 |
| 20 to <25% | 6084 (37.9) | 6046 (37.6) | 0.005 | 10,162 (37.6) | 14,714 (39.0) | 0.028 |
| 25 to <30% | 3104 (19.3) | 3088 (19.2) | 0.003 | 5099 (18.9) | 7634 (20.3) | 0.034 |
| ≥30% | 297 (1.9) | 313 (2) | 0.007 | 434 (1.6) | 974 (2.6) | 0.068 |
| Other laboratory investigations during baseline period | | | | | | |
| LDL Cholesterol, mg per dL | 86.8 (24.02) | 86.8 (23.5) | 0.001 | 85.9 (23.6) | 88.0 (24.3) | 0.91 |
| HDL, mg per dL | 38.9 (9.1) | 38.9 (9.1) | 0.002 | 38.2 (8.8) | 39.9 (9.7) | 0.188 |
| Total Cholesterol, mg per dL | 160.8 (31.5) | 160.6 (31.7) | 0.006 | 160.5 (31.5) | 161 (31.6) | 0.015 |
| eGFR, mL per minute per 1.73 m$^2$ | 48.5 (11.7) | 48.5 (11.8) | 0.003 | 49 (11.5) | 47.7 (11.8) | 0.111 |
| Creatinine, mg per dL | 1.65 (0.7) | 1.66 (0.8) | 0.007 | 1.63 (0.67) | 1.68 (0.83) | 0.66 |
| Other medications utilized during baseline period | | | | | | |
| ACEi/ARB | 14,718 (91.6) | 14,723 (91.6) | 0.001 | 25,082 (92.9) | 33,593 (89.1) | 0.134 |
| Alzheimer meds | 563 (3.5) | 594 (3.7) | 0.01 | 796 (3.0) | 1691 (4.5) | 0.081 |
| Anti-anginal | 5167 (32.1) | 5076 (31.6) | 0.012 | 9686 (35.9) | 10234 (27.1) | 0.189 |
| Anti-arrhythmic | 1638 (10.2) | 1611 (10.0) | 0.006 | 3014 (11.2) | 3461 (9.2) | 0.066 |
| Oral Anticoagulant | 3588 (22.3) | 3574 (22.2) | 0.002 | 6559 (24.3) | 71,234 (18.9) | 0.131 |
| Parenteral anticoagulant | 1444 (9) | 1409 (8.8) | 0.008 | 2633 (9.8) | 2782 (7.4) | 0.085 |
| Antidepressant | 8652 (53.8) | 8719 (54.2) | 0.008 | 15,419 (57.1) | 17,432 (46.2) | 0.219 |
| Other antihypertensives | 7603 (47.3) | 7651 (47.6) | 0.006 | 13,020 (48.2) | 17,094 (45.3) | 0.058 |
| Other antiplatelets | 4744 (29.5) | 4641 (28.9) | 0.014 | 8629 (32.0) | 9808 (26.0) | 0.131 |
| Antipsychotic | 1769 (11) | 1796 (11.2) | 0.005 | 3022 (11.2) | 3704 (9.8) | 0.045 |
| Anti-smoking | 3882 (24.2) | 3966 (24.7) | 0.012 | 7122 (26.4) | 7487 (19.9) | 0.155 |
| ASA | 8465 (52.7) | 8510 (52.9) | 0.006 | 14,721 (54.5) | 17,746 (47.1) | 0.15 |
| Beta blocker | 11,932 (74.2) | 11,913 (74.1) | 0.003 | 20,983 (77.7) | 26,065 (69.1) | 0.196 |
| Benzodiazepine | 4768 (29.7) | 4795 (29.8) | 0.004 | 8453 (31.3) | 9829 (26.1) | 0.116 |
| Calcium channel blocker | 10,622 (66.1) | 10,640 (66.2) | 0.002 | 18,198 (67.4) | 23,694 (62.8) | 0.096 |
| COPD medications | 4147 (25.8) | 4161 (25.9) | 0.002 | 7656 (28.4) | 8419 (22.3) | 0.139 |
| Corticosteroids | 5651 (35.2) | 5662 (35.2) | 0.001 | 10,108 (37.4) | 11,363 (30.1) | 0.155 |
| Diuretics | 10,884 (67.7) | 10,875 (67.7) | 0.001 | 18,876 (69.9) | 23,809 (63.1) | 0.144 |
| Loop diuretics | 7923 (49.3) | 7953 (49.5) | 0.004 | 14,885 (55.1) | 16,039 (42.5) | 0.254 |
| Non-statin | 6543 (40.7) | 6482 (40.3) | 0.008 | 11,774 (43.6) | 13,388 (35.5) | 0.166 |
| Statin | 14,976 (93.2) | 14,971 (93.1) | 0.001 | 25,559 (94.7) | 33,956 (90.1) | 0.175 |
| CKD stage at index date | | | | | | |
| Creatinine, mg per dL | 2.0 (1.1) | 2.0 (1.1) | 0.008 | 2.0 (1.0) | 2.0 (1.1) | 0.053 |
| eGFR, mL per minute per 1.73 m$^2$ | 37.5 (11.1) | 37.5 (11.4) | 0.001 | 37.3 (10.8) | 37.9 (11.3) | 0.053 |
| CKD stage 1 | 13 (0.08) | 12 (0.07) | 0.002 | 19 (0.1) | 35 (0.1) | 0.008 |
| CKD stage 2 | 379 (2.4) | 398 (2.48) | 0.008 | 599 (2.2) | 914 (2.4) | 0.014 |
| CKD stage 3a | 2756 (17.1) | 2786 (17.3) | 0.005 | 4459 (16.5) | 6654 (17.7) | 0.03 |
| CKD stage 3b | 9486 (59) | 9384 (58.4) | 0.13 | 15,883 (58.8) | 22,691 (60.2) | 0.027 |
| CKD stage 4 | 2870 (17.9) | 2897 (18) | 0.004 | 5287 (19.6) | 5911 (15.7) | 0.103 |
| CKD stage 5 | 572 (3.6) | 599 (3.7) | 0.009 | 750 (2.8) | 1503 (4.0) | 0.067 |
| Glycated hemoglobin at index date | | | | | | |
| Glycated hemoglobin at index date | 8.5 (1.6) | 8.5 (1.7) | 0.001 | 8.6 (1.6) | 8.1 (1.6) | 0.32 |
| ≤7.5% | 4484 (27.9) | 4502 (28) | 0.003 | 6322 (23.4) | 14,233 (37.8) | 0.31 |
| 7.5% to <9.0% | 6425 (40) | 6385 (39.7) | 0.005 | 10,836 (40.1) | 14,332 (38.0) | 0.04 |
| ≥9.0% | 5167 (32.1) | 5189 (32.3) | 0.003 | 9839 (36.4) | 9143 (24.3) | 0.27 |

Data are mean (SD) or numbers (%) unless stated otherwise.

*ACEi/ARB* angiotensin-converting enzyme inhibitors or angiotensin II receptor blockers, *CABG* coronary artery bypass graft, *COPD* chronic obstructive pulmonary disease, *EPY* event per patient per year, *HDL* high density lipoprotein, *LDL* low density lipoprotein, *PCI* percutaneous coronary intervention, *PS* propensity score, *SD* standard deviation, *SGLT2i* sodium glucose cotransporter 2 inhibitors, *St diff* standardized difference, *GLP1-RA* glucagon like peptide 1 receptor agonists, *DPP4i* dipeptidyl peptidase 4 inhibitors.

**Table 2 | Analysis of primary (acute healthcare utilization), secondary (all-cause death and composite cardiovascular events), and exploratory (combined renal events) outcomes of the propensity score-matched cohort**

| Outcome | GLP1-RA<br>n = 16,076 | DPP4i<br>n = 16,076 | Odds ratio or coefficient of regression (95%CI) | p-value |
|---|---|---|---|---|
| Annual Rate of acute healthcare utilization, EPY | 1.52 (4.8) | 1.67 (4.4) | −0.15 (−0.25 to −0.05)[a] | 0.004 |
| All-cause Death | 2847 (17.7%) | 3287 (20.5%) | 0.84 (0.79 to 0.89)[b] | <0.001 |
| Cardiovascular events | 1757 (10.9%) | 1782 (11.1%) | 0.98 (0.92 to 1.06)[b] | 0.66 |
| Combined kidney outcome | 359 (2.23%) | 557 (3.46%) | 0.64 (0.56 to 0.73)[b] | <0.001 |

Data are mean (SD) or numbers (%) unless stated otherwise. Multivariate linear regression analysis was performed for annual rate of healthcare utilization. Logistic regression analysis performed for all other outcomes, adjusting for propensity score. Each outcome was assessed in a separate model where the outcome was the dependent variable and use of GLP1-RA or DPP4i as predictor variable.
[a]Coefficient of regression.
[b]Odds ratio; EPY, event per patient per year; GLP1-RA, glucagon like peptide 1 receptor agonists; DPP4i, dipeptidyl peptidase 4 inhibitors.

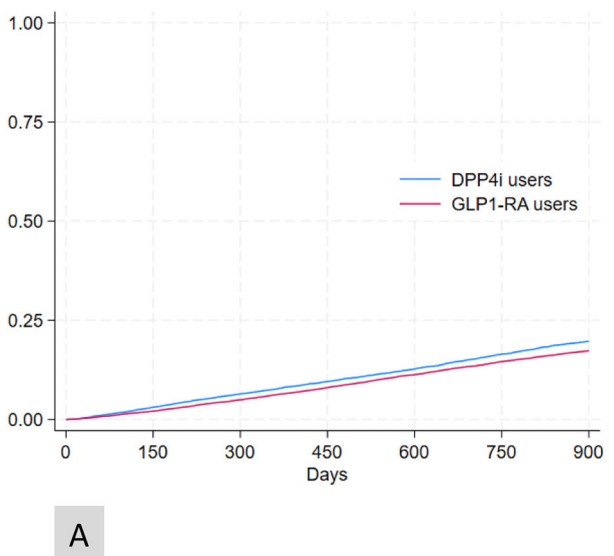
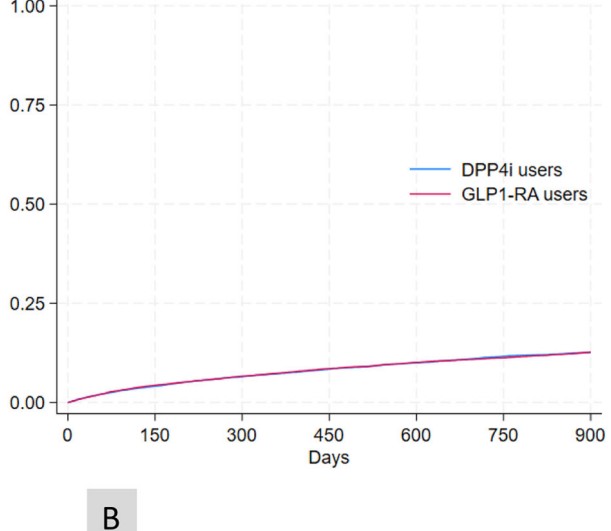

**Fig. 2 | Kaplan-Meier curves for all-cause mortality and first composite cardiovascular event.** Panel 2A: Kaplan–Meier curves for time to all-cause mortality (HR: 0.86, 95%CI: 0.81–0.90); and Panel 2B: Kaplan–Meier curves for cardiovascular events (HR: 0.99, 95%CI: 0.93–1.06). CI confidence interval, DPP4i dipeptidase 4 inhibitors, GLP1-RA glucagon like peptide 1 receptor agonists, HR hazard ratio, K-M Kaplan-Meier failure estimates.

95%CI 1.05–1.21). Gastrointestinal symptoms were not significantly different in GLP1-RA users compared to DPP4i users (23.3% vs 22.7%, respectively; OR 1.04, 95%CI 0.98–1.09) (Supplementary Table S4 in supplementary online material).

Exploratory subgroup analysis of the whole cohort showed similar overall trends in benefits in rates of acute healthcare utilization, all-cause mortality, and CKD progression but not combined cardiovascular composite event outcome within the GLP1-RA group compared to DPP4i group, although there was treatment effect heterogeneity between several subgroups (Supplementary Fig. S3 [A-D] in supplementary online material). In the post-hoc per protocol analysis, a total of 16,075 pairs of propensity score-matched patients were included. There were no differences in any baseline characteristics between groups, after matching. Outcomes are illustrated in Supplementary Table S5 and Supplementary Fig. S4.

## Discussion

The current study, emulating a hypothetical clinical trial, reveals significantly lower acute healthcare utilization among a national cohort of patients enrolled at the VHA with moderate to advanced CKD following initiation of GLP1-RA as compared to propensity score matched active comparators. Over a mean (SD) follow-up of 2.2 (1.9) years, the annual rate of acute healthcare utilization was 10% lower in the GLP1-RA group compared to active comparators. Additionally, our study found that the odds of death from any cause was 16% lower in the GLP1-

RA group compared to active comparators within the same follow-up period. No difference was found in a composite of cardiovascular event outcomes (which did not include death). The odds of having CKD progression was 36% lower within the GLP1-RA group as compared to DPP4i group.

Our population of patients with moderate to advanced CKD constituted older patients (mean age 72 years) with high prevalence of diabetes complications (over 50% with at least one complication), high frailty (~60% had moderate or severe frailty and <2% were not frail), high comorbidity burden (mean weighted CCI score of 7.3), and high mortality (~19% during follow up period). Furthermore, this population had high utilization of medications known to improve outcomes (for example, 93% were on statins, 92% were on Angiotensin-converting enzyme inhibitors or angiotensin II receptor blockers, among others). Therefore, the noted improvements in healthcare utilization, mortality, and kidney outcomes are in addition to what could be provided by the best-known standard of care and therefore clinically relevant.

Previous studies in a general population with T2D have demonstrated cost savings with use of sodium glucose transporter-2 inhibitor (SGLT2i) as compared to either DDP4i or GLP1-RA[22–25], however, there have been no prior studies to our knowledge examining direct cross-comparison of healthcare utilization between use of DDP4i and GLP1-RA in CKD in the United States. Incretin mimetics have demonstrated safety and efficacy in advanced kidney disease, however multiple studies have shown that use of GLP1-RA have shown improvements in

glycemic control, body weight reductions, improved beta cell function, and decreased incidence hypoglycemia as compared to DPP4i in a general diabetes population[26–31], effects which might explain the reduction of acute healthcare utilization observed in this study of patients with CKD treated with GLP1-RA as compared to DPP4i.

Cardiovascular outcomes studies in people with T2D and high cardiovascular risk have shown that DPP4i have an overall cardio-neutral profile[32–37], while GLP1-RAs were associated with decreased all-cause mortality and major adverse cardiovascular events (MACE)[38]. However, not all studies evaluating use of GLP1-RAs on CV outcomes have shown significant reductions in non-fatal CV events. For example, ELIXA trial found no difference in the primary endpoint of composite CV events (13.4% within the treatment group with lixisenatide compared to 13.2% within the placebo group)[39]. Additionally, PIONEER 6 study, there was no noted difference in the occurrence of non-fatal myocardial infarction, non-fatal stroke, or unstable angina with oral semaglutide[40]. Furthermore, most evidence for cardiovascular event reduction with use of GLP1-RA has been within a general diabetes population, and evidence for patients with moderate to advanced kidney disease has been limited. In a pooled metanalysis of major placebo-controlled trials, Sattar et al found a 12% reduction in all-cause mortality and 14% relative risk reduction in 3-point MACE; there was no statistical heterogeneity in subgroup analysis by eGFR, however, the proportion of patients with CKD was low (only 14% in GLP-1RA group had eGFR < 60 mL per min per 1.73 m$^2$)[21]. Studies that have included larger proportions of patients with CKD are few, and the observations on cardiovascular outcomes have been mixed. For example, in a separate pooled analysis of 4 major clinical trials reporting cardiovascular events in patients with T2D and CKD, Kelly et al. failed to find an association between GLP-1RA and reduction of composite cardiovascular events[41]. Recently, several retrospective studies of patients with T2D and CKD found that although use of GLP1-RA was associated with improved overall survival compared to use of DPP4i, there was no significant difference in cardiovascular outcomes[42,43]. Our finding of similar incidence of non-fatal cardiovascular outcomes with GLP1-RA compared to DPP4i is in line with the recently published FLOW study that randomized people with early to mid-stage CKD to semaglutide or placebo, which reported similar incidence of non-fatal myocardial infarction and non-fatal stroke across the semaglutide and placebo treated groups, despite a significantly reduced risk of major kidney disease events and all-cause death[44].

Our finding of a significant reduction in CKD progression outcome in association with GLP1-RA use is also consistent with the findings of the FLOW study[44]. This effect on CKD progression persisted across all subgroups, although there was some heterogeneity of effect based on index date. Regardless, our study provides real-world evidence in support of the renal-protective effect of GLP1-RA on kidney function in patients with moderate to advanced CKD, although mechanisms of kidney protection remain to be elucidated. The effect of GLP1-RA on kidney function is likely multifactorial – possible indirect factors such as effect on weight reduction and glycemic control might contribute, in addition to direct effects on inflammation, oxidative stress, and natriuresis[45–47].

## Limitations
Our study has several limitations. First, due to the retrospective nature of our study, we were unable to verify medication adherence. We defined index date as the date of first pharmacy fill of either GLP1-RA or DPP4i, and operated under the assumption that patients are taking the prescribed medication as directed. Secondly, there is heterogeneity within medication classes and doses; different GLP1-RA were prescribed throughout the study according to the GLP1-RA agent approved within the VHA at time of prescription (Supplemental Table S6). Thirdly, the class of GLP1-RA medications are restricted within the VHA system and require pharmacy approval or

endocrinology consultation[48], which may have resulted in potential heterogeneity between the GLP1-RA and DPP4i group. As such, we designed this study using propensity score matching method and included as many covariates as possible. Fourth, our data cannot identify cause of death; hence, we could not verify if the decrease in mortality was related to a decrease in cardiovascular mortality or other disease categories. Similarly, our definition for cardiovascular diseases did not include death due to acute cardiovascular events. Lastly, our study cohort was predominantly Caucasian and male, reflective of the population cared for within the VHA system, thus results may not be generalizable to a wider population, although some recent studies found that VHA population have similar health characteristics as individuals with other insurance coverage suggesting greater generalizability.

In conclusion, this real-world study emulating a target clinical trial shows that among a national cohort of patients enrolled at the VHA with moderate to advanced CKD, use of GLP1-RA was associated with lower annual rate of acute healthcare utilization, lower all-cause mortality, and lower kidney events as compared to treatment with DPP4i. There was no significant difference in cardiovascular events (not including CV death) between the matched groups. Further studies are needed to validate these findings and to elucidate the mechanisms of clinically important outcomes with GLP1-RA use in patients with moderate to advanced CKD.

## Methods
This retrospective cohort study was approved by the Orlando Veterans Affairs Health Care System Institutional Review Boards (protocol number 1680940-1), which waived informed consent since data were fully de-identified before providing access to the investigators. This study followed Reporting of Observational Studies in Epidemiology (STROBE) guidelines. This study used national data from the VHA Corporate Data Warehouse hosted at VHA Informatics and Computing Infrastructure (VINCI) to emulate a hypothetical target clinical trial. VINCI is a research platform for health services research at VHA according to published and validated protocols[49]. We extracted data from fiscal years (FY) 2006 to end of FY 2021 from patients who met eligibility criteria and filled either GLP1-RA or DPP4i prescriptions. Extracted data encompassed inpatient and outpatient diagnosis and procedure codes, vital signs, laboratory data, and pharmacy fill data.

### Study population
The study included all adults aged 35 years or older who are regular VHA users (as defined by presence of at least one inpatient or outpatient medical encounter), with at least one set of vital signs, and one set of laboratory investigation (including glucose, glycated hemoglobin [HbA1c], serum creatinine, and lipid panel) during the baseline period. From this population, we identified a cohort of patients with moderate to advanced kidney disease defined as having two or more consecutive eGFR values < 45 mL per min per 1.73 m$^2$ obtained over a span of three consecutive months and who were newly initiated on GLP1-RA or DPP4i at a time having already met the eGFR criteria. Cohort entry was defined as the date the above eGFR criteria was met. Diabetes was not a specified inclusion criterion; however, VA formulary at time of the study restricted use of these medications to patients with T2D[48].

### Study groups
We used an active comparator, new-user design to emulate a clinical trial, in which participants are newly initiated on study's medications. A summary of the protocol emulating a randomized control trial is outlined in the supplementary table S7. This design mitigates the risk of immortal time bias and minimize confounding due to unmeasured characteristics, as previously described[50]. We also used propensity

score to match study groups on predefined characteristics, emulating participants' randomizations using predefined stratifications criteria The study groups included: (1) GLP1-RA group: patients who initiated GLP1-RA, and (2) active comparator group, consisted of patients who initiated DPP4i, both initiated after meeting eGFR entry criteria. We excluded prevalent users who initiated the respective treatment before cohort entry and patients who were concomitant users of GLP1-RA and DPP4i. Entries were censored at the last date of the study period, or the date of GLP1-RA initiation among DDP4i users who discontinued DDP4i and started GLP1-RA. We also censored entries at time of death when estimating the acute cardiovascular composite event outcome.

## Study intervals

Index date was defined as the date of initiation (first fill) of either GLP1-RA or DPP4i. The baseline period, used to describe baseline characteristics of the study groups, included the period between the first available inpatient or outpatient encounter during the study period (FY2006–FY2021) and the index date. The follow up period used to examine outcomes irrespective of ongoing use of the medication (emulating a modified intention-to-treat analysis of a clinical trial), started from the index date, and continued until study end (October 30, 2021), death, or initiation of a GLP1-RA in the DPP4i group, whichever came first (Fig. S1 in Supplementary online Methods).

## Pre-specified outcomes

The primary outcome was defined as the annual rate of acute health-care utilization (number of events per year of follow-up): calculated as the sum of urgent care visits, Emergency Department visits, and hospitalizations divided by the duration of follow-up. Urgent care or ED visits were identified by stop codes as described in Managerial Cost Accounting Office of the VHA (see list in Supplementary Table S1)[51].

The pre-specified secondary outcomes were (1) all cause-mortality and (2) incidence of a cardiovascular composite event outcome that included: first occurrence of acute myocardial infarction, cardiac arrest or ventricular fibrillation, acute stroke, or coronary revascularization, as previously described and validated using validated International Classification of Diseases, 9th Revision, Clinical Modification (ICD-9-CM) and 10th Revision, Clinical Modification (ICD-10-CM) and procedure codes[52–54]. One inpatient diagnosis, or 2 different outpatient encounter diagnoses were required to meet criteria for a respective event (see list in Supplementary Table S9).We extracted date of death from the VHA Death Ascertainment File which contain mortality data from the Master Person Index file in CDW and the Social Security Administration Death Master File.

## Post-hoc outcomes

We examined the following outcomes:

1. Composite outcome of CKD progression: defined as doubling of serum creatinine during follow up (mean serum creatinine during follow up divided by mean serum creatinine at index date ≥2) or incident stage 5 CKD during follow up period. Albuminuria during follow-up period was also examined, but not included in the composite outcome due to high percentage of missing values.
2. Safety outcomes: examined odds of hypoglycemic events using administrative codes validated in prior studies[55,56] and gastro-intestinal symptoms (constipation, diarrhea, change in bowel habits, abdominal pain, or ileus) during follow up period using administrative codes, as described in prior publications[57].

Prespecified subgroup analyses were stratified by duration of GLP1-RA or DPP4i use (at least 6-months, 1-year, and 2-years of use) within their respective groups; and stratified by CKD disease stage at index date (stage 3a or better, stage 3b, and stage 4 or worse). Further post-hoc exploratory subgroup analyses of outcomes within the whole cohort (before propensity score-matching) were completed within the following subgroups: (1) age (less than 60 years of age), 60 to 75 years, and greater than 75 years, (2) body mass index (less than 30 kg per m$^2$, between 30 to 45 kg per m$^2$, and above 45 kg per m$^2$), (3) frailty status at baseline (non-frail or frail), (4) Charlson comorbidity index (low CCI, moderate CCI, or high CCI), (5) change in weight at follow-up from baseline (no weight loss or weight gain, less than 5% weight loss, between 5 and <10% weight loss, or greater than 10% weight loss), (6) index date (before or after 10/1/2015), and (7) excluding patients with eGFR <15 mL per min per 1.73 m$^2$. We also examined our outcomes after adjusting for mean hemoglobin A1c during follow up, use of other classes of glucose-lowering medications, and number of glucose-lowering medication classes used during follow up period. This last analysis was added to explore if outcomes might be related to other medications added during follow up, rather than GLP1-RA or DPP4i use.

## Data extraction

This study used extracted data from the national VINCI database. We extracted data from fiscal years (FY) 2006 to end of FY 2021 from patients who met eligibility criteria. Laboratory data were extracted at two different points: throughout the baseline period and at the point closest to the index date, since the long baseline period may not reflect the actual value at time of drug initiation. Laboratory values were captured using 2 different techniques: (1) searching the name of the test in the laboratory database and (2) using LOINC codes that were known to be at utilization in the VA at its time (supplemental table S10). Duplicate values were identified and removed. Extreme values were discarded. Overall, only 0.21% of the laboratory investigation values were identified as extreme and discarded. Missing data was replaced by the mean when possible (see supplemental Table S11)

eGFR was calculated using MDRD formula without race consideration: GFR, in mL/min per 1.73 m$^2$ = 175 × SCr (exp[−1.154]) × Age (exp[−0.203]) × (0.742 if female). Stage 5 CKD was defined as an incident decrease in mean estimated glomerular filtration rate (eGFR) during the last year of follow up to <15 mL per min per 1.73 m$^2$ (stage 5). Similarly, extreme values for weight, systolic blood pressure, and diastolic blood pressure were discarded. Overall, 0.0007% of the values were identified as extreme and discarded and, imputed values constituted <0.02% of available data.

## Cohort characterization and propensity score matching

Patients' baseline demographic, clinical characteristics, and healthcare utilization were extracted from the entire baseline period. Disease categories were described using validated definitions[58–60]. To ensure comparability of the two treatment groups, we calculated 3 different comorbidity scores: the VHA frailty index, using a previously described approach[60]; the weighted Charlson Comorbidity Index (CCI) which have been shown to improve the performance over the original score with c-statistics of 0.8[61,62]; and cardiovascular risk as calculated by D'Agostino method[63]. We created a propensity score to match GLP1-RA users and DDP4i users at a ratio of 1:1 using 120 characteristics selected a priori (listed in Table 1), Variables included: age, gender (self-reported), race and ethnicity (self-reported), demographics, personal history, vital signs, comorbidities, comorbidity and cardiovascular scores, frailty score, healthcare utilization, laboratory values, glucose-lowering medication classes, and other medication classes.

We used the routine by Leuven and Sianesi to perform nearest number matching using the logit model with no replacement[64,65]. We explored a caliper width of 0.01, which approximately represented 0.2 times the standard deviation of the logit of the propensity scores, as suggested in prior publications[66]. This caliper achieved balance in differences, without residual statistically significant differences

between treatment groups on all covariates. After propensity score creation, pseudo R2 decreased to <0.0001, indicating that successful balance has been achieved[67]. Supplemental Figs. S2(A) and S2(B) depict kernel graphs of propensity score before and after matching, respectively.

## Statistical analyses

Statistical analyses were performed using STATA version 18 (College Station, TX) via the secured VINCI workspace. Baseline characteristics were compared using Chi-square tests for dichotomous variables and *t*-test for continuous variables. The primary analysis examined the difference in annual rate of acute healthcare utilization within the propensity score matched cohort using *t*-test and multivariate linear regression analysis. The secondary outcomes (all-cause mortality and cardiovascular composite event outcome) and post-hoc outcomes (combined kidney outcome and safety outcomes) were compared using logistic regression analysis, in which the outcome of interest was the dependent variable and use of GLP-RA or DDP41 as independent variable. Statistical significance of primary and secondary outcomes was defined as two-tailed *p*-values < 0.05. We also performed time to event analysis using Cox proportional-hazards regression model to estimate the hazard ratio (HR) and 95% confidence intervals (CI) for all-cause mortality and cardiovascular composite event outcome for GLP1-RA group compared to DDP4i group. Entries were censored at the last date of the study period, or the date of GLP1-RA initiation among DDP4i users who discontinued DDP4i and started GLP1-RA. We also censored entries in date of death, when estimating the cardiovascular composite event outcome. Additional exploratory subgroup analysis of the overall cohort was performed with separate regression models with outcome of interest being the dependent variable, use of GLP1-RA or DDP4i as independent variable, and propensity score as a covariate. Subgroup analysis was conducted using the overall cohort to maximize sample size within each subgroup. Interaction terms with variables considered in subgroups analysis were also examined. Since 40 separate interaction analyses were performed, an adjusted *p*-value of *p* < 0.0012 indicates statistically significant interaction after applying Bonferroni correction.

Lastly, we conducted a post-hoc per-protocol analysis, in which outcomes were truncated or censored at 90 days after the date of the last prescription of either medication in their perspective group. To avoid ascertainment bias due to differences in duration of medication use in the two comparison groups, we created a new propensity score-matched cohort including the duration of follow-up based on per-protocol analysis, using the same previous specifications of the primary analysis. We used the same baseline demographics, clinical characteristics, and healthcare utilization used in the primary analysis. Similar to our primary analysis, we calculated 3 different comorbidity scores: the VHA frailty index[60], the weighted CCI[61,62],and cardiovascular risk as calculated by D'Agostino method[63]. We created a propensity score, using 120 characteristics, as in primary analysis except for duration of follow up, which reflected now the duration of follow up based on per-protocol analysis. We matched the comparison groups using a multivariable logistic regression to estimate the propensity score and performed nearest number matching with a caliper of 0.01 with no replacement, using same technique as our primary analysis. This caliper was found to reduce standardized differences after matching to <0.1 and maximize study size.

## Reporting summary

Further information on research design is available in the Nature Portfolio Reporting Summary linked to this article.

## Data availability

Data used in this study reside in the VA Informatics and Computing Infrastructure (VINCI), the operational platform for health services research at the Veterans Healthcare Administration (VHA). VINCI acts as data steward for VHA Data Systems. By VHA Office of Research and Development mandates, VINCI does not allow the copying, transferring, or printing of any data out of its secure environment, except in aggregate format. Access to data by other researchers requires official employment with the VHA. For more information on VINCI protocols, please refer to the VHA website: https://www.research.va.gov/programs/vinci (or http://vaww.vinci.med.va.gov/VinciCentral if accessing via VHA intranet).

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

## Acknowledgements

This material is the result of work supported with resources and the use of facilities at the Orlando VA Healthcare System (Orlando, Florida). The views expressed herein are those of the authors and do not reflect the official policy or position of the Veterans Health Administration, or the US Government or any of its affiliated entities. Some of the authors are employees of the US government. This work was prepared as part of their official duties and, as such, there is no copyright to be transferred. MK participation in this research was supported (in whole or in part) by HCA Healthcare and/or an HCA Healthcare affiliated entity. The views expressed in this publication represent those of the author(s) and do not necessarily represent the official views of HCA Healthcare or any of its affiliated entities.

## Author contributions

F.S., S.Z., I.A.M., M.K., I.L. performed initial conceptualization and design of the project. I.A.M., C.A.A., M.K., I.L., S.Z. contributed to data curation and methodology. I.A.M. performed formal statistical analysis using STATA. S.Z. consolidated data analysis, created visualizations, and drafted the initial manuscript. All authors discussed the results and contributed to the review of the final manuscript.

## Competing interests

I.L. received research funding (paid to institution) from NovoNordisk, Sanofi, Merck, Pfizer, Mylan, Boehringer-Ingelheim. IL received advisory/consulting fees and/or other support from: Novo Nordisk, Eli Lilly, Sanofi, Astra Zeneca, Boehringer-Ingelheim, Cytoki Pharma, Johnson and Johnson, Intercept, TARGETPharma, Merck, Pfizer, Valeritas, Zealand Pharma, Shionogi, Carmot Therapeutics, Structure Therapeutics, Bayer, Translational Medical Academy, Mediflix, Biomea, Metsera, Regeneron, The Comm Group, and WebMD. I.L. serves on the Data Safety Monitoring Board for JAEB. C.A.A. received research funding (paid to the institution) from Merck, Bristol Myers Squibb, and Boehringer-Ingelheim. C.A.A. received funding from the National Institutes of Health National Center for Advancing Translational Sciences (grant #UL1TR003163). S.Z., F.S and I.A.M. declared no conflict of interest.
