## [Transparent Peer Review file · Nature Communications]

Healthcare Utilization, Mortality, and Cardiovascular Events Following GLP1-RA Initiation in Chronic Kidney Disease

Corresponding Author: Dr Ishak Mansi

Version 0:

Reviewer comments:

Reviewer #1

(Remarks to the Author)

To compare outcomes following initiation of GLP1-RA in patients with T2D and advanced CKD the authors performed a retrospective propensity score-matched cohort study among Veterans during fiscal years 2006 to 2021 and compared the results to Dipeptidyl peptidase-4 inhibitors (DPP4i). Primary outcome was rate of acute healthcare utilization. Secondary outcomes were all-cause mortality and a composite of acute cardiovascular events (not including death).

The eligible cohort included 26,997 GLP1-RA users and 37,708 DPP4 users. After propensity score matching (16,076 pairs) and 2.2 years mean follow-up duration, use of GLP1-RA was associated with lower rate of acute healthcare utilization (1.52 events/year for GLP1-RA vs 1.64 in DPP4i; coefficient of regression $\beta = -0.15$, 95% CI -0.25 to -0.05, $p=0.004$) and lower all-cause mortality as compared to the DPP4i group (17.7% in GLP1-RA vs 20.5% in DPP4i; OR 0.84, 95% CI 0.79 to 0.89, $p<0.01$). There was no significant difference in acute cardiovascular events.

The authors conclude that the use of GLP1-RA in patients with advanced CKD was associated with lower annual rate of acute healthcare utilization and all-cause mortality. There was no significant difference in acute cardiovascular events between the matched groups.

The subject of the study is important, and the sample size is large. The propensity matching is meticulous with nearly perfect matching. Some concerns emerge.

1. Most GLP1RA trials have demonstrated protection from atherosclerotic cardiovascular events. (eg PMID 36508493). The authors report no effect on CV events. Perhaps their coding is too noisy to detect benefit that were not hard to demonstrate in RCTs. For example, they define cv composite as the first occurrence of acute myocardial infarction, cardiac arrest or ventricular fibrillation, acute stroke, or coronary revascularization. The trials define this as cv death (which the authors note is not captured in their database), non-fatal MI and non-fatal stroke. Furthermore, it is unclear if the veterans were receiving care outside the VA and some events were missed. It is hard to explain a benefit on all cause mortality with no effect on cardiovascular outcomes.

2. The authors say that they are studying advanced CKD. That is typically defined at eGFR <30 . If so, Table 1 suggests that only about 20% of the patients had advanced CKD. If they excluded patients with stage 1 and 2 CKD, then we could study moderate to advanced CKD (that is eGFR <60) and that would be reasonable, but we cannot say advanced CKD and then have only 20% of the patients with advanced CKD in the sample.

3. The above comment is particularly relevant in the context of power calculations provided by the authors: "Database has around 150,000 GLP1 users and 250,000 DPP4 users. Assuming that 20% have advanced CKD, we will have around 30,000 GLP1 users and 50,000 DPP4 users. assuming we will be able to match 50% of subjects, we will end up with about 15,000 subjects in each PS matched arm. Assuming that GLP1 will lower rate of healthcare use by 10%, the study will have power $>99\%$ to detect difference at an alfa of 0.05" Clearly the assumptions for advanced CKD are not met therefore, the study is underpowered per se.

4. In my view, the authors should have excluded those on dialysis. These patients have a completely different trajectory compared to those not on dialysis.

5. Table 2 is difficult to interpret. The results column (coefficient of regression or ratio) is a mixed bag of outcomes, eg hazard ratio, odds ratio, coefficient of regression. It could be done better.

6. Table 3 shows effect modification by duration of medication use or CKD stage. It is typically depicted using a forest plot. Furthermore, the reader is left wondering if there is or whether there is no effect modification. For this, one would have to report the p value of the interaction.

7. There is plenty of lost opportunity in the cohort assembled. Why did not the authors not study effect modification by baseline body mass index? Was medication possession ratio associated with an improved outcome? Or was weight loss associated with better outcomes? Could adherence be studied here as a way to explain the positive results? There are multiple markers of social determinants of health. Given that all have equal access to healthcare in the VA system, the social determinants modifying the effect could be an important question.

8. Finally, the FLOW trial was recently stopped because of efficacy. The inclusion of ESKD as an endpoint in this cohort could have meaningfully provided information relevant to the broader community.

Reviewer #2

(Remarks to the Author)

1. This real world comparative study conducted by Shuyao Zhang et.al regarding on the effects/potential advantages of GLP-1RA initiation(compared to DPP4i) in patients with T2D and advanced CKD did show noteworthy results which included lower rates of acute healthcare utilization and all cause mortality. These findings add valuable knowledge not only extend the scope of benefits beyond organ protection of GLP-1 RA but also expand the beneficent to advanced DKD.

2.Previous studies of long acting GLP-1 RA including CVOTs, post-hoc analysis ,meta-analysis or real world studies showed effects of significant CV risk reduction and potential renal protection, especially for albuminuria reduction. The majority of patients enrolled in these studies have normal or mildly impaired-range eGFR and normal albuminuria. The evidence of their acting effects on advanced DKD is still lacking or insufficient. This study helps fill in some of the missing evidence and can facilitate the conduct of further research.

3.This research was done very rigorously and utilized many adjusting/corrective methods to avoid bias. I believe the work meets the expected standards and supports the conclusions.

Minor revision:

4. However, there are still several shortages that need to be clarified at this time. Firstly, the causes of acute health care utilization were not reported in this study. As we all know, DPP4i causes less hypoglycemia and GI tract intolerance than GLP-1RA especially in patients with advanced DKD. To analyze the incidence of ER visiting for hypoglycemia or GI symptoms related side effects(dehydration, electrolyte imbalance, or acute kidney injury etc) between the two groups is very important and the result can further strengthen the safety of using GLP-1RA in patients with advanced DKD. Secondly, the achieved HbA1c and its difference between two groups were not disclosed in this study. It is very important to know that suboptimal HbA1c control will lead to increase the rate of acute healthcare utilization and the possibility of adding more anti-glycemic drugs is likely higher in DPP4i group. If this is the case, the add-on SU, TZD or insulin therapy may contribute to more hypoglycemia or drug-related complications(BW gain/edema / or heart failure etc.) in DPP4i group. To justify this bias/influential effects is very important to support the real extra-benefits of GLP-1RA .

Major revision:

Thirdly, the most of all, GLP-1 RA has shown renal protection in terms of albuminuria reduction and slowing down eGFR decline(in KDIGO guideline). Several post hoc analysis of GLP-1 RA CVOTs also suggest patients with macroalbuminuria-DKD benefited more from GLP-1RA therapy. FLOW trial stopped early due to evidence of renal protection with semaglutide and the preliminary result has confirmed this viewpoint that use of semaglutide 1.0 mg (Ozempic) was associated with a 24% reduction in risk of kidney disease-related events among people with type 2 diabetes and chronic kidney disease (CKD) with median UACR of 567mg/g(68.5% with macroalbuminuria) . Unfortunately, in current study, no data of albuminuria and percentage distribution of albuminuria categories in two groups were recorded. If the albuminuria levels are averaged to be in macroalbuminuria level, patients treated with GLP-1RA will benefit more in reducing rate of renal endpoints(eGFR declined, or doubling, ESRD or renal death). The more rapid deterioration of renal function was reported to cause more acute healthcare utilization, CV event/death and all cause mortality. Thus, albuminuria distribution report and subsequential renal endpoints/outcome analysis should be done in this study and all the data will make the conclusions more credible and robust.

Reviewer #3

(Remarks to the Author)

1. In this study, study participants were selected from 2006 to 2021, which covers 15 years; however, the mean follow-up time was only 26.3 months (2.2±1.9 years). The follow-up time in the Kaplan-Meier curves as shown in Fig 2 also ended at 900 days (<3 years). As shown in Table 1, those who were recently recruited (about 94% of participants were recruited after 2016 and about 67% after 2019) could significantly dominate the study results. I suggest the study participants could be classified into two subgroups: before 2015 and after 2016. The former subgroup, although accounts for only 5%, could demonstrate long-term (at least 5 years) effects of using GLP-1RA compared to DPP4i.

2. The authors defined regular VHA users (lines 133-134) as "presence of at least one inpatient or outpatient medical encounter." I wonder how many participants who visited VHA for only one inpatient or outpatient. In literature, the so-called

“regular users of a certain medical facility” were usually defined by “at least 2 or 3 inpatient or outpatient medical encounters within one year.” Please clarify this definition.

3. In the paragraph of “study population,” the authors indicated the eligibility of the study participants was their eGFR<45; however, about 20% of the study participants were at CKD stage 1-3a (Table 1), indicating their eGFR>45, contradictory to the inclusion criteria. Why?

4. In lines 149-151, the authors detailed the statistical methods: “Entries were censored at the last date of the study period, or the date of GLP1-RA initiation among DPP4i users who discontinued DPP4i and started GLP1-RA.” In addition to that, how did the authors handle the scenario that GLP1-RA users who discontinued GLP1-RA and started DPP4i? Furthermore, how to handle the situation that patients stopped taking both of GLP1-RA and DPP4i after initiating either medication for a certain period of time?

5. For the multivariable analyses including linear regression, logistic regression, and Cox model, what covariates were adjusted in the propensity score matched models? Who did you handle the non-independent condition for the PPS matched cohort?

6. To show robustness of the study results, I encourage the authors conducting more subgroups analyses for this vulnerable group: for example, stratification by age (e.g., <60 vs. ≥60), frailty status, and CCI level.

7. If the “real-world comparative study” was emphasized as indicated in the title, the concept of this study was to use observational data to evaluate the clinical effectiveness of an intervention (which was GLP1-RA use in this study). For this type of research question, using a “target trial emulation approach” may be a better study design because by this approach we could make observational study to be the “equivalent” randomized clinical trial that enhances evidence level for the “real-world comparative study.” The authors may find it useful to follow the framework described by Hernan et al (Am J Epidemiology 2016;183:758-64; J Clin Epidemiol 2016;79:70-75).

Version 1:

Reviewer comments:

Reviewer #1

(Remarks to the Author)

Thank you

Reviewer #2

(Remarks to the Author)

All my suggested minor revisions have been well addressed point by point in the revised version. The authors also added a new post-hoc outcome: composite CKD progression outcome, which is closely relevant to the risks of acute healthcare utilization, CV event/death and all cause mortality

Unfortunately, for my major concerns regarding on the mean level of albuminuria and percentage distribution of albuminuria categories in both groups remain undocumented. In the CVOTs of cornerstone pharmacotherapies of DKD(e.g., ACEi/ARB, SGLT2i or NSMRA), the beneficial effects of individual treatment all showed a dose dependent relationship between albuminuria reduction and the outcome. Thus, the study should conduct albuminuria analysis as described above and these data will make the conclusions more credible and robust.

Reviewer #3

(Remarks to the Author)

I am satisfied with most of the answers to my inquiries except the only application of an intention-to-treat manner to emulate a randomized control trial. In a long follow-up observational cohort study, it is not uncommon to observe a treatment switch in either arm. It may not be realistic to believe GLP-1RA could still have its residual effects 3 years after the medication discontinuation. The authors should show a frequency table (or figure) delineating the treatment shift (from GLP-1RA yes to no; from GLP-1RA no to yes; from DPP4i yes to no; and from DPP4i no to yes) across the entire observational period. The authors are also encouraged to apply an as-treated method, or time-varying methodology to ensure robustness of their study results derived from the intention-to-treat method.

Version 2:

Reviewer comments:

Reviewer #2

(Remarks to the Author)

The authors' reply highlighted that less than 60% of the cohort had an albuminuria measurement during follow-up, which might introduce bias in the outcome assessment. They explained the possible reasons for the missing albuminuria data and added Supplemental Table 4 to describe the baseline levels and progression of albuminuria between groups.

I acknowledge the challenges in collecting complete lab data in cohort studies and would like to propose one final minor revision to the authors: In Supplemental Table 4, the term ‘microalbumin in urine’ should be corrected to ‘albumin level in urine’. “Progression of microalbuminuria” should be corrected to “progression of albuminuria”. The category of “>= 3 to <300” should be corrected to “>= 30 to <300”

Reviewer #3

(Remarks to the Author)

The authors have adequately responded to my suggestions. No further questions.

Dear Reviewers:

We would like to share our deep appreciation for your efforts in reviewing our paper and providing valuable comments and feedback. Below we provide the point-by-point responses. All modifications in the manuscript are in track changes for ease of reviewing.

Respectfully, on behalf of all authors

Ishak Mansi, MD (corresponding author)

Reviewer #1 (Remarks to the Author):	Authors' responses:
To compare outcomes following initiation of GLP1-RA in patients with T2D and advanced CKD the authors performed a retrospective propensity score-matched cohort study among Veterans during fiscal years 2006 to 2021 and compared the results to Dipeptidyl peptidase-4 inhibitors (DPP4i). Primary outcome was rate of acute healthcare utilization. Secondary outcomes were all-cause mortality and a composite of acute cardiovascular events (not including death). The eligible cohort included 26,997 GLP1-RA users and 37,708 DPP4 users. After propensity score matching (16,076 pairs) and 2.2 years mean follow-up duration, use of GLP1-RA was associated with lower rate of acute healthcare utilization (1.52 events/year for GLP1-RA	We shared the reviewer's surprise when we first noted these results. However, there are several noteworthy points that likely explain these findings. First, there is no universal definition for components of cardiovascular (CV) events and different studies used different definitions.¹ As the reviewer points out, CV death is not included in our CV event endpoint. This is because there is no reliable way to identify CV death from medical records; in the cardiovascular outcome trials this endpoint is always adjudicated and even with adjudication it is often a difficult endpoint to accurately ascertain. Aside this practical limitation, we believe our CV endpoint provides important insights into the effects of these drugs in this population with chronic kidney disease (CKD). Second, while it is possible that some non-fatal CV events were not captured due to care for such episodes being rendered exclusively outside of the VA system, such occurrence should be equally distributed (non-differential) across the two groups and will not have a large influence on the overall results. Third, our finding of lack of cardiovascular benefit in patients with chronic kidney disease has been noted in other observational studies. For example, a recent study of 8922 patients with advanced diabetic kidney disease noted that composite cardiovascular events in the GLP-1RA and DPP-4i groups were not different (HR:

vs 1.64 in DPP4i; coefficient of regression $\beta = -0.15$, 95% CI -0.25 to -0.05, $p=0.004$) and lower all-cause mortality as compared to the DPP4i group (17.7% in GLP1-RA vs 20.5% in DPP4i; OR 0.84, 95% CI 0.79 to 0.89, $p<0.01$). There was no significant difference in acute cardiovascular events.

The authors conclude that the use of GLP1-RA in patients with advanced CKD was associated with lower annual rate of acute healthcare utilization and all-cause mortality. There was no significant difference in acute cardiovascular events between the matched groups.

The subject of the study is important, and the sample size is large. The propensity matching is meticulous with nearly perfect matching. Some concerns emerge.

1. Most GLP1RA trials have demonstrated protection from atherosclerotic cardiovascular events. (eg PMID 36508493). The authors report no effect on CV events. Perhaps their coding is too noisy to detect benefit that were not hard to demonstrate in RCTs. For example, they define cv composite as the first occurrence of acute myocardial infarction, cardiac arrest or ventricular fibrillation, acute stroke, or coronary revascularization. The trials define this as cv

0.88, 95% CI 0.68–1.13).² Another recent study, of 27,279 patients with diabetes and advanced CKD, reported that GLP-1RA compared to DPP4i was associated with lower all-cause mortality (HR: 0.79; 95%CI: 0.63-0.98) but similar CV outcome (HR: 1.05, 95%CT: 0.77-1.44).³

Fourth, among randomized controlled trials (RCT), not all RCTs evaluating CV outcomes have shown significant reduction in non-fatal CV events with GLP1-RA. For example, in the PIONEER 6 study, there was no noted difference in the occurrence of non-fatal myocardial infarction, non-fatal stroke, or unstable angina.⁴ More importantly, in the recently published FLOW study, which evaluated semaglutide vs placebo in a population with early to mid-stage CKD, similar findings were noted: the risk of non-fatal myocardial infarction and non-fatal stroke were comparable between the semaglutide and placebo treated groups, while all-cause death was significantly reduced in the semaglutide treated group compared to placebo.⁵ In a meta-analysis of 4 CV outcome trials that included patients with CKD (as defined by $eGFR<60$), Kelly et al. found no difference in CV outcomes (defined by cardiovascular death, nonfatal myocardial infarction, and nonfatal stroke).⁶

However, we thank the reviewer for bringing up this important point, since many readers may wonder about the same. Therefore, we expanded on these explanations in the discussion section to further contextualize these results for the reader (please see page 16, 1st paragraph).

death (which the authors note is not captured in their database), non-fatal MI and non-fatal stroke. Furthermore, it is unclear if the veterans were receiving care outside the VA and some events were missed. It is hard to explain a benefit on all cause mortality with no effect on cardiovascular outcomes.	
2. The authors say that they are studying advanced CKD. That is typically defined at eGFR <30. If so, Table 1 suggests that only about 20% of the patients had advanced CKD. If they excluded patients with stage 1 and 2 CKD, then we could study moderate to advanced CKD (that is eGFR <60) and that would be reasonable, but we cannot say advanced CKD and then have only 20% of the patients with advanced CKD in the sample.	Thank you for this comment. Cohort entry was restricted to patients who had at least two consecutive eGFR values <45 over a span of 3 months prior to the index date. However, the duration of the baseline period, which extended from cohort entry till medication initiation was long (on average 10 years) and some fluctuation of eGFR values over time is expected. At index date, the average eGFR in the propensity matched cohort was 37.5, and >80% of the cohort met criteria for CKD stage 3b or worse at index date. However, we agree with the reviewer that it would be more accurate to describe the cohort as having moderate to advanced CKD and therefore we revised accordingly throughout the manuscript.
3. The above comment is particularly relevant in the context of power calculations provided by the authors: "Database has around 150,000 GLP1 users and 250,000 DPP4 users. Assuming that 20% have advanced CKD, we will have around 30,000 GLP1 users and 50,000 DPP4 users. assuming we will be able to match 50% of subjects, we will end up with about 15,000 subjects in each PS matched arm. Assuming that GLP1 will lower rate of healthcare use by 10%, the study will have power >99% to detect difference at an alfa of 0.05" Clearly the	For this analysis, we defined our cohort of "advanced CKD" as those with two consecutive eGFR values <45 during the baseline period. We agree that the terminology used is confusing to the reader, therefore we updated the manuscript throughout to reflect the fact that we are studying a population with moderate to advanced CKD.

assumptions for advanced CKD are not met therefore, the study is underpowered per se.	
4. In my view, the authors should have excluded those on dialysis. These patients have a completely different trajectory compared to those not on dialysis.	Although we agree that this population might indeed have a different trajectory, patients treated with dialysis have high healthcare utilization and mortality, so it is very relevant to examine the effect of GLP-1RAs on these outcomes in this population. GLP-1RA and DPP4i are approved for use in patients with CKD including patients with ESRD on dialysis, and GLP-1RAs are recommended in this population for their cardioprotective effects. For these reasons our pre-specified analysis includes this subgroup. However, we agree that it is important to also explore the subgroup without ESRD, so we conducted an additional exploratory analysis excluding subjects with eGFR <15 mL/min per 1.73 m². The results of this new analysis were similar to those of the primary analysis, and this new analysis is now reported in the supplemental materials (Supplemental Table S4).
5. Table 2 is difficult to interpret. The results column (coefficient of regression or ratio) is a mixed bag of outcomes, eg hazard ratio, odds ratio, coefficient of regression. It could be done better.	Thank you for this feedback. We have edited Table 2 to make it easier to interpret.
6. Table 3 shows effect modification by duration of medication use or CKD stage. It is typically depicted using a forest plot. Furthermore, the reader is left wondering if there is or whether there is no effect modification. For this, one would have to report the p value of the interaction.	We thank the reviewer for this comment; we have updated these results (along with additional subgroup analysis requested by other reviewers) in a forest plot. The data from prior Table 3 along with additional subgroup analyses are now reported in the supplemental materials (Supplementary Figure S3).
7. There is plenty of lost opportunity in the cohort assembled. Why did not the authors not study effect modification by baseline body mass index? Was medication	We thank the reviewer for these suggestions which we found very helpful. We added all the suggested additional analysis which are now reported in the Forest Plot (Supplemental Figure S3) and in the supplementary file.

possession ratio associated with an improved outcome? Or was weight loss associated with better outcomes? Could adherence be studied here to explain the positive results? There are multiple markers of social determinants of health. Given that all have equal access to healthcare in the VA system, the social determinants modifying the effect could be an important question.	Due to equal access of care and medications within the VA system, cost and access to medication is equal. Social determinants of health, which include income level, depravity level, education level, rural residence, service connection, marital status, are very important. The association of these factors with our outcomes of interest, though relevant, is beyond this study's scope. A separate dedicated study would be needed to explore to what degree different social determinants might impact the effects of GLP-1 RAs on health in this population.
8. Finally, the FLOW trial was recently stopped because of efficacy. The inclusion of ESKD as an endpoint in this cohort could have meaningfully provided information relevant to the broader community.	We appreciate the reviewer's recommendation. While this endpoint was not within the original scope of our work and therefore is not pre-specified, we took on the valuable advice. We created a new composite outcome of CKD progression defined as doubling of serum creatinine during follow up or incident stage 5 CKD during follow up. These results are now presented in the manuscript and importantly, are aligned with those reported now in the FLOW trial. We have added the new "Composite outcome of CKD progression" as post-hoc outcome in the Method section (page 9) and the results were added to all tables. Additional discussion of these results and those of the FLOW trial has now been added to the manuscript in the Discussion section (page 18, second and third paragraphs).

Reviewer #2 comments:	Authors' responses:
1. This real world comparative study conducted by Shuyao Zhang et.al regarding on the effects/potential advantages of GLP-1RA iniiation(compared to DPP4i) in patients with T2D and advanced CKD did show noteworthy results which included lower rates of acute healthcare utilization and all cause mortality. These findings add valuable	Thank you very much for acknowledging the impact of our work on the current understanding of the effects of the GLP-1 RA class on relevant clinical outcomes in the population of interest with moderate-to-advanced CKD.

knowledge not only extend the scope of benefits beyond organ protection of GLP-1 RA but also expand the beneficent to advanced DKD.	
2.Previous studies of long acting GLP-1 RA including CVOTs, post-hoc analysis ,meta-analysis or real world studies showed effects of significant CV risk reduction and potential renal protection, especially for albuminuria reduction. The majority of patients enrolled in these studies have normal or mildly impaired-range eGFR and normal albuminuria. The evidence of their acting effects on advanced DKD is still lacking or insufficient. This study helps fill in some of the missing evidence and can facilitate the conduct of further research.	Thank you for noting that the population we studied is hugely underrepresented in the existing literature and therefore represents a knowledge gap.
3.This research was done very rigorously and utilized many adjusting/corrective methods to avoid bias. I believe the work meets the expected standards and supports the conclusions.	Thank you for appreciating the detailed and rigorous nature of the work we performed.
Minor revision: 4. However, there are still several shortages that need to be clarified at this time. Firstly, the causes of acute health care utilization were not reported in this study. As we all know, DPP4i causes less hypoglycemia and GI tract intolerance than GLP-1RA especially in patients with advanced DKD. To analyze the incidence of ER visiting for hypoglycemia or	Thank you for these suggestions. We agree that adding safety outcomes are important and relevant. We added two additional safety outcomes: incidence of hypoglycemia and GI-related symptoms using validated measures, as post-hoc outcomes (Method section, page 9). Hypoglycemic events were more common in GLP1-RA group (OR: 1.13; 95%CI 1.05-1.21). GI-related symptoms were very prevalent in both groups (around one fourth of patients) but was not significantly different (OR: 1.04; 95%CI: 0.98-1.09) between groups. We reported the results in Table S6 in supplement. We also agree that analysis of reasons for decrease in acute care utilization is very important, however, the reasons for acute care

GI symptoms related side effects(dehydration, electrolyte imbalance, or acute kidney injury etc) between the two groups is very important and the result can further strengthen the safety of using GLP-1RA in patients with advanced DKD. Secondly, the achieved HbA1c and its difference between two groups were not disclosed in this study. It is very important to know that suboptimal HbA1c control will lead to increase the rate of acute healthcare utilization and the possibility of adding more anti-glycemic drugs is likely higher in DPP4i group. If this is the case, the add-on SU, TZD or insulin therapy may contribute to more hypoglycemia or drug-related complications(BW gain/edema / or heart failure etc.) in DPP4i group. To justify this bias/influential effects is very important to support the real extra-benefits of GLP-1RA.	utilization include a wide range of diseases beyond this study's scope and should be a subject of further studies. The finding that CKD progression outcome (now added based on reviewers' comments), which was significantly lower in GLP1-RA users, may lend an explanation. The reviewer's second point is very relevant and intends to differentiate whether the noted benefits in this study are due to the use of GLP-1RA or rather due to harm from using add-on glucose lowering medications in the DPP-4i group. To address this, we have completed two additional analyses:  1. We compared average HbA1c during follow-up between the two treatment groups and found no statistically significant difference. Furthermore, the proportion of patients with a HbA1c>6.5% or >9% was similar between treatment groups. 2. We repeated the analysis in PS-cohort adjusting for the average HbA1c during the follow up period, number of antidiabetes medications used during follow up, and use of the following classes of meds: Glitazones, Glinides, Insulins, Metformin, SGLT2, and Sulfonylurea during the follow up period. Our results remained consistent for all outcomes. We have added to the Method section (last line in Page 9 and first 2 lines in page 10) a description for this analysis. Results are now presented in the supplementary file (Supplementary Table S5).
Major revision: Thirdly, the most of all, GLP-1 RA has shown renal protection in terms of albuminuria reduction and slowing down eGFR decline(in KDIGO guideline). Several post hoc analysis of GLP-1 RA CVOTs also suggest patients with macroalbuminuria-DKD benefited more from GLP-1RA therapy. FLOW trial stopped early due to evidence of renal protection with	Thank you for this valuable comment. We have now added a new post-hoc outcome: composite CKD progression outcome; Please see full response to the similar comment provided by reviewer #1. Our findings are very consistent with the FLOW study.

semaglutide and the preliminary result has confirmed this viewpoint that use of semaglutide 1.0 mg (Ozempic) was associated with a 24% reduction in risk of kidney disease-related events among people with type 2 diabetes and chronic kidney disease (CKD) with median UACR of 567mg/g(68.5% with macroalbuminuria) . Unfortunately, in current study, no data of albuminuria and percentage distribution of albuminuria categories in two groups were recorded. If the albuminuria levels are averaged to be in macroalbuminuria level, patients treated with GLP-1RA will benefit more in reducing rate of renal endpoints(eGFR declined, cr doubling, ESRD or renal death). The more rapid deterioration of renal function was reported to cause more acute healthcare utilization, CV event/death and all cause mortality. Thus, albuminuria distribution report and subsequential renal endpoints/outcome analysis should be done in this study and all the data will make the conclusions more credible and robust.	
Reviewer #3 comments:	Authors' responses:
1. In this study, study participants were selected from 2006 to 2021, which covers 15 years; however, the mean follow-up time was only 26.3 months (2.2±1.9 years). The follow-up time in the Kaplan-Meier curves as shown in Fig 2 also ended at 900 days (<3	Thank you for this suggestion. Within the VHA system patients are typically followed longitudinally for long periods of time, which is a major strength to our study. Given the RCTs that demonstrated the beneficial cardiometabolic effects of GLP1-RA, their utilization had exponentially increased. We agree that the duration of drug use might impact the outcomes analyzed, therefore we conducted a sensitivity analysis by duration of drug use, results of which supported the primary

years). As shown in Table 1, those who were recently recruited (about 94% of participants were recruited after 2016 and about 67% after 2019) could significantly dominate the study results. I suggest the study participants could be classified into two subgroups: before 2015 and after 2016. The former subgroup, although accounts for only 5%, could demonstrate long-term (at least 5 years) effects of using GLP1-RA compared to DPP4i.	analysis. We also conducted the requested analysis by the time of index date (pre and post 2015). Of note, the pre-2015 cohort was very small, especially in the GLP-1RA group, but the overall trend of the results was similar to that in the primary analysis. This exploratory analysis has been added to supplemental figure S3.
2. The authors defined regular VHA users (lines 133-134) as “presence of at least one inpatient or outpatient medical encounter.” I wonder how many participants who visited VHA for only one inpatient or outpatient. In literature, the so-called “regular users of a certain medical facility” were usually defined by “at least 2 or 3 inpatient or outpatient medical encounters within one year.” Please clarify this definition.	There is no standard definition for VHA users and different studies used different definitions. We defined regular VHA users as having all of the following: (1) at least 1 healthcare visit; (2) at least one set of vital signs; and (3) at least one set of specific laboratory data. Since few patients may use VHA system to only dispense medications for their low fixed copayments within the VHA, we preferred not to use number of visits alone as criteria for being VHA user, but rather to include vital signs and laboratory investigations. Our criteria was successfully used in prior publications.⁷ Most importantly, table 1 shows that our population had extensively used the VHA system with >300 outpatient visits at baseline and that utilization was similar in both comparison groups in the propensity score matched cohort.
3. In the paragraph of “study population,” the authors indicated the eligibility of the study participants was their eGFR<45; however, about 20% of the study participants were at CKD stage 1-3a (Table 1), indicating their eGFR>45, contradictory to the inclusion criteria. Why?	Patients are followed longitudinally at the VHA for long time periods, whereas GLP1-RA and DPP4i are relatively newer medications (or their use only recently increased). Patients entered our cohorts when they had at least two consecutive eGFR values <45 that were at least 3-months apart (cohort entry), but many years could have passed from this timepoint until they were initiated on GLP1-RA or DPP4i (period from cohort entry to index date was occasionally very long). During this time eGFR can change, either from random fluctuation or because of other interventions. For this reason, 20% of those who met eligibility criteria had an eGFR>45 at the time of the index date. However, it is important to note that the

	average eGFR at index date was 37.5, which therefore indicates that we are reporting on a cohort of patients with much more advanced CKD compared to other cohorts previously reported on. Upon Reviewer's #1 suggestion, we clarified across the manuscript that this is a cohort with moderate-to-advanced CKD.
4. In lines 149-151, the authors detailed the statistical methods: "Entries were censored at the last date of the study period, or the date of GLP1-RA initiation among DPP4i users who discontinued DPP4i and started GLP1-RA." In addition to that, how did the authors handle the scenario that GLP1-RA users who discontinued GLP1-RA and started DPP4i? Furthermore, how to handle the situation that patients stopped taking both of GLP1-RA and DPP4i after initiating either medication for a certain period of time?	Thank you for your question. We intended to treat the cohort in a similar manner to a randomized trial and therefore we applied an intention-to-treat-like principle to our analysis. Furthermore, we treated DPP4i as placebo (emulating a clinical trial). Therefore, if a patient switched treatment from a GLP-1RA to a DPP4i, they were analyzed in the GLP-1RA group, regardless of medication discontinuation. In both groups, patients were followed irrespective of treatment discontinuation.
5. For the multivariable analyses including linear regression, logistic regression, and Cox model, what covariates were adjusted in the propensity score matched models? Who did you handle the non-independent condition for the PPS matched cohort?	We did not adjust for any variables in the propensity score matched cohort, since patients were perfectly matched without any residual imbalances. For analysis including the overall cohort (before matching), we adjusted to the propensity score. These details of statistical analysis are reported in Supplementary Methods (to comply with the journal word count limit). Since other readers may have the same question, we added to the legend of table 2 the following statement: "Each outcome was assessed in a separate model where the outcome was the dependent variable and use of GLP1-RA or DPP4i as predictor variable." For supplemental figure S3, we added to the Figure legend the following statement: "Analyses are adjusted for propensity score."
6. To show robustness of the study results, I encourage the authors conducting more subgroups analyses for this vulnerable group: for example, stratification by age (e.g., <60 vs. >=60), frailty status, and CCI level.	We thank the reviewer for the suggestions. We conducted the additional analyses which were all in line with the primary analysis (albeit some of the subgroups were small and thus not powered to show a statistically significant difference). These results of these additional analyses are now reported in Supplemental Figure S3.

7. If the “real-world comparative study” was emphasized as indicated in the title, the concept of this study was to use observational data to evaluate the clinical effectiveness of an intervention (which was GLP1-RA use in this study). For this type of research question, using a “target trial emulation approach” may be a better study design because by this approach we could make observational study to be the "equivalent" randomized clinical trial that enhances evidence level for the “real-world comparative study.” The authors may find it useful to follow the framework described by Hernan et al (Am J Epidemiology 2016;183:758-64; J Clin Epidemiol 2016;79:70-75).	Thank you for this comment. Our goal with propensity-score matching was to emulate a randomized trial using observational data. We have created a table in the supplemental appendix to reflect the protocol components of our emulation (Supplementary appendix section A).
--	---

References:

- 1 Kip, K. E., Hollabaugh, K., Marroquin, O. C. & Williams, D. O. The problem with composite end points in cardiovascular studies: the story of major adverse cardiac events and percutaneous coronary intervention. *J Am Coll Cardiol* **51**, 701-707 (2008). [https://doi.org/S0735-1097\(07\)03694-7](https://doi.org/S0735-1097(07)03694-7) [pii]10.1016/j.jacc.2007.10.034
- 2 Lin, Y. *et al.* The cardiovascular and renal effects of glucagon-like peptide 1 receptor agonists in patients with advanced diabetic kidney disease. *Cardiovascular Diabetology* **22**, 60 (2023). <https://doi.org/10.1186/s12933-023-01793-9>
- 3 Chen, J. J. *et al.* Association of Glucagon-Like Peptide-1 Receptor Agonist vs Dipeptidyl Peptidase-4 Inhibitor Use With Mortality Among Patients With Type 2 Diabetes and Advanced Chronic Kidney Disease. *JAMA Netw Open* **5**, e221169 (2022). <https://doi.org/10.1001/jamanetworkopen.2022.1169>
- 4 Husain, M. *et al.* Oral Semaglutide and Cardiovascular Outcomes in Patients with Type 2 Diabetes. *N Engl J Med* **381**, 841-851 (2019). <https://doi.org/10.1056/NEJMoa1901118>
- 5 Perkovic, V. *et al.* Effects of Semaglutide on Chronic Kidney Disease in Patients with Type 2 Diabetes. *N Engl J Med* **391**, 109-121 (2024). <https://doi.org/10.1056/NEJMoa2403347>
- 6 Kelly, M. *et al.* Effects of GLP-1 receptor agonists on cardiovascular outcomes in patients with type 2 diabetes and chronic kidney disease: A systematic review and meta-analysis. *Pharmacotherapy* **42**, 921-928 (2022). <https://doi.org/10.1002/phar.2737>
- 7 Mansi, I. A. *et al.* Association of Statin Therapy Initiation With Diabetes Progression: A Retrospective Matched-Cohort Study. *JAMA Intern Med* **181**, 1562-1574 (2021). <https://doi.org/10.1001/jamainternmed.2021.5714>

Dear reviewers:

We would like to express our sincere appreciation for the effort you and the reviewers have put into evaluating our manuscript and subsequent revisions. We have carefully considered the additional comments and addressed the responses as below.

Respectfully,

Corresponding Author;
Ishak A. Mansi, MD
Education Services, Orlando VA Medical Center;
Professor, Department of Medicine,
University of Central Florida, Orlando, FL
13800 Veterans Way, Orlando FL 32827

Reviewer #1 (Remarks to the Author):

Thank you

Reviewer #2 (Remarks to the Author):

All my suggested minor revisions have been well addressed point by point in the revised version. The authors also added a new post-hoc outcome: composite CKD progression outcome, which is closely relevant to the risks of acute healthcare utilization, CV event/death and all cause mortality

Unfortunately, for my major concerns regarding on the mean level of albuminuria and percentage distribution of albuminuria categories in both groups remain undocumented. In the CVOTs of cornerstone pharmacotherapies of DKD (e.g., ACEi/ARB, SGLT2i or NSMRA), the beneficial effects of individual treatment all showed a dose dependent relationship between albuminuria reduction and the outcome. Thus, the study should conduct albuminuria analysis as described above and these data will make the conclusions more credible and robust.

Response: Thank you very much for your feedback. We agree that measurement of urinary albumin excretion is an important surrogate marker for kidney disease. This marker is frequently used as an end point (by itself or as part of a composite) in kidney-oriented randomized clinical trials (RCTs) that are smaller and of shorter duration and therefore do not have the power to detect differences in hard kidney endpoints. In RCTs, urine samples are collected prospectively and systematically, and even under these conditions there is a fair

amount of data missingness. Measurements of urine albumin creatinine ratio (ACR) in the real world are unfortunately not performed consistently. In a recent meta-analysis, only 35.1% of 1,303,027 patients with diabetes underwent screening ACR.¹ A CDC report noted that the percentage of Veterans receiving albuminuria testing was 15.4% in 2022.² In our data, less than 60% of the cohort had an albuminuria measurement during follow up and therefore we are concerned that including an outcome with this level of missingness (which is likely not at random) will induce bias in outcome assessment.

However, we agree that an analysis of change in albuminuria would be a meaningful addition to our reporting, and therefore, in response to the reviewer's comment, we added a Supplemental Table S8 which describes the progression of microalbuminuria during follow up in our study population.

Reviewer #3 (Remarks to the Author):

I am satisfied with most of the answers to my inquiries except the only application of an intention-to-treat manner to emulate a randomized control trial. In a long follow-up observational cohort study, it is not uncommon to observe a treatment switch in either arm. It may not be realistic to believe GLP-1RA could still have its residual effects 3 years after the medication discontinuation. The authors should show a frequency table (or figure) delineating the treatment shift (from GLP-1RA yes to no; from GLP-1RA no to yes; from DPP4i yes to no; and from DPP4i no to yes) across the entire observational period. The authors are also encouraged to apply an as-treated method, or time-varying methodology to ensure robustness of their study results derived from the intention-to-treat method.

Response: In response to the reviewer's suggestion, we performed a post-hoc per-protocol analysis, we created another propensity score matched cohort that incorporated duration of follow up, to mitigate ascertainment bias. We briefly described our procedure and results for this cohort in the Methods and the Results sections of the manuscript and detailed them in Supplemental Method section H, Supplemental Table S8, and Supplemental Figure S4. We also added to the Result section additional information about the duration of use of either study medications and proportion of patients who discontinued or switched the respective medication during the follow-up period (first paragraph in Result section and Supplemental Figure 1).

References:

1. Shin JI, Chang AR, Grams ME, et al. Albuminuria Testing in Hypertension and Diabetes: An Individual-Participant Data Meta-Analysis in a Global Consortium. *Hypertension*. Sep 2021;78(4):1042-1052. doi:10.1161/hypertensionaha.121.17323
2. . *Albuminuria Testing among US Veterans Kidney Disease Surveillance System; Quality of Care Centers for Disease Control and Prevention Available at: <https://nccdcdc.gov/CKD/detail.aspx?Qnum=Q640&Strat=Diabetes#refreshPosition> (last accessed August 27, 2024).*

Response to reviewers (revision 3)

First, we ask you to revise your paper to address our editorial requests (in the attached Author Checklist) and any remaining comments from reviewers (included at the end of this email, if applicable).

Response: We have followed the instructions in the Author Checklist and completed the form. Thank you.

REVIEWERS' COMMENTS

Reviewer #2 (Remarks to the Author):

The authors' reply highlighted that less than 60% of the cohort had an albuminuria measurement during follow-up, which might introduce bias in the outcome assessment. They explained the possible reasons for the missing albuminuria data and added Supplemental Table 4 to describe the baseline levels and progression of albuminuria between groups.

I acknowledge the challenges in collecting complete lab data in cohort studies and would like to propose one final minor revision to the authors: In Supplemental Table 4, the term 'microalbumin in urine' should be corrected to 'albumin level in urine'. "Progression of microalbuminuria" should be corrected to "progression of albuminuria". The category of " ≥ 3 to <300 " should be corrected to " ≥ 30 to <300 "

Response: We thank the reviewer; the corrections were completed.

Reviewer #3 (Remarks to the Author):

The authors have adequately responded to my suggestions. No further questions.

Response: We thank the reviewer for the effort and time in reviewing our manuscript.